# Emerging Applications of Stereotactic Ablative Radiotherapy in Oligometastatic Colorectal Cancer

**DOI:** 10.3390/ijms262110302

**Published:** 2025-10-23

**Authors:** Hasan Al-Sattar, Esele Okondo, Amir Mashia Jaafari, Inesh Sood, Jakob Hassan Dinif, Su Yin Lim, Charlotte Hafkamp, Irene Chong, Joao R. Galante, Sola Adeleke

**Affiliations:** 1School of Medicine and Biomedical Sciences, University of Oxford, Oxford OX1 3PL, UK; hasan.al-sattar@magd.ox.ac.uk (H.A.-S.); esele.okondo@magd.ox.ac.uk (E.O.); amir.jaafari@trinity.ox.ac.uk (A.M.J.); inesh.sood@worc.ox.ac.uk (I.S.); 2School of Medicine, St George’s University of London, London SW17 0RE, UK; m2100221@sgul.ac.uk; 3Radiotherapy Department, Royal Marsden Hospital, London SW3 6JZ, UK; suyin.lim@rmh.nhs.uk (S.Y.L.); charlotte.hafkamp@rmh.nhs.uk (C.H.); irene.chong@rmh.nhs.uk (I.C.); joao.galante@rmh.nhs.uk (J.R.G.); 4Amsterdam University Medical Centre, 1105 AZ Amsterdam, The Netherlands; 5School of Biomedical Engineering and Imaging Sciences, King’s College London, London WC2R 2LS, UK

**Keywords:** stereotactic ablative radiotherapy, SABR, stereotactic body radiotherapy, SBRT, colorectal cancer, oligometastases, immunotherapy, artificial intelligence

## Abstract

Colorectal cancer (CRC) is a major cause of cancer mortality worldwide, with metastatic disease remaining the main driver of poor prognosis. In recent years, the concept of oligometastatic disease, where patients present with a limited number of metastases, has created an opportunity to use local therapies with curative intent. Stereotactic ablative radiotherapy (SABR) has become increasingly important in this setting, as it allows the delivery of high, ablative doses with excellent local control and generally low toxicity. Notably, randomised data such as SABR-COMET, alongside large prospective series including SABR-5, have demonstrated improvements in survival outcomes in the context of oligometastatic disease across mixed primary tumour types, with CRC patients making up a relatively small proportion in these trials. This has presented SABR as a practical treatment option for patients with oligometastatic CRC, although more CRC-specific phase III trials are needed. Other challenges include the radioresistance of CRC metastases, and treatment outcomes that vary depending on the anatomical site, tumour biology, and prior therapies. Technical issues such as motion management and organ-at-risk constraints also continue to limit dose escalation. Emerging strategies—including MR-guided radiotherapy, proton-based SABR, integration with systemic agents such as immunotherapy, and the use of biomarkers and artificial intelligence to refine patient selection—are beginning to address these limitations. This review summarises the current evidence and emerging advancements to highlight how SABR may evolve as part of an integrated approach to oligometastatic CRC.

## 1. Introduction

Colorectal cancer (CRC) remains a leading cause of cancer-related mortality, being the fourth deadliest cancer globally, and only second to lung cancer in the EU and UK [1,2,3,4]. This can primarily be attributed to metastatic disease, which significantly reduces overall survival (OS) and quality of life [5,6,7]. Historically, the development of metastatic lesions has indicated a shift from curative to palliative care; however, recent advancements in the understanding and management of metastatic CRC have led to the recognition of the oligometastatic disease state [8,9,10]. This intermediate clinical stage, characterised by a limited number of metastatic lesions (typically 1 to 5), provides a unique opportunity for aggressive localised interventions aimed at improving survival outcomes [11,12,13,14]. Among the treatment options available for oligometastatic CRC, stereotactic ablative radiotherapy (SABR), also known as stereotactic body radiotherapy (SBRT), has emerged as a promising therapeutic option [15,16]. SABR delivers highly targeted, ablative doses of radiation to metastatic lesions, resulting in excellent local control (LC), while sparing the surrounding healthy tissue and minimising toxicity [17]. The ability to administer a high biological effective dose (BED) has translated into improved OS, progression-free survival (PFS), and LC compared to conventional radiotherapy techniques, as evidenced by landmark trials such as SABR-COMET, SABR-5, and ORCHESTRA [18,19,20]. These favourable outcomes have propelled SABR into routine clinical practice for patients with limited metastatic disease across various malignancies, including colorectal cancer.

Despite its demonstrated efficacy, however, the application of SABR in oligometastatic CRC presents unique challenges, driven primarily by tumour heterogeneity, variable radiosensitivity, and technical considerations such as motion management [21,22]. Colorectal cancer metastases are notably radioresistant compared to metastases from other primary tumour sites, necessitating tailored treatment strategies involving higher radiation doses or combination therapies [22,23,24,25]. Furthermore, the accurate delivery of SABR remains challenging due to respiratory-induced tumour motion, geometric uncertainties, toxicity to gastrointestinal organs at risk (OARs), and the limitations associated with current imaging technologies [26,27]. To overcome these barriers, guidelines recommend motion management techniques such as abdominal compression, active breathing control, gating, and sophisticated immobilisation and imaging methods, highlighting a clear need for continual technological innovation [28,29,30].

An emerging body of evidence underscores the importance of precise patient selection, suggesting that the anatomical site, genomic factors (e.g., KRAS and TP53 mutations), primary tumour origin (colon versus rectum), and clinical variables significantly influence treatment outcomes [31,32,33]. Identifying patients who will derive maximal benefits from SABR is therefore crucial to improving prognostic accuracy and treatment efficacy. Additionally, advances in imaging and radiotherapy technology, including MR-guided radiotherapy, proton-based SABR, and real-time adaptive strategies, are increasingly enhancing treatment precision and safety, further optimising clinical outcomes [34,35,36,37]. Moreover, recent research highlights the potential synergy between SABR and systemic therapies, particularly immunotherapy [38]. The preliminary data indicate that combining SABR with immune checkpoint inhibitors may enhance anti-tumour immune responses, potentially overcoming the resistance observed in microsatellite stable colorectal cancers. Concurrently, artificial intelligence (AI) and machine learning approaches for SABR, although still emerging, have the potential to facilitate precision medicine through improved treatment planning, predictive modelling, and personalised therapeutic strategies [39].

This review will evaluate the current practices and the literature for SABR in oligometastatic CRC, highlighting some if its current limitations. We will highlight emerging advancements and innovative strategies in SABR to overcome the latter, emphasising a multidisciplinary and integrated approach to enhance patient outcomes.

## 2. Results

### 2.1. Current Outlook

SABR has become a treatment modality for oligometastatic CRC in recent years due to its ability to precisely deliver a high BED10 (as an alpha/beta ratio of 10 is conventionally used for clinical SBRT planning purposes) to the metastatic sites in the body with limited effects on the adjacent organs at risk (OAR). The SABR-COMET trial demonstrates the effectiveness of SABR as a treatment, with a 5-year OS rate of 42.3% with SABR compared to 17.7% with standard treatment [18]. The study further found the PFS to be 17.3% with SABR compared to 3.2%, with no significant increase in toxicity. However, adverse events were reported to be higher in the SABR group, with 29% experiencing grade 2 or greater side effects compared to 9% in the control group. Additionally, only 18 out of the 99 patients in this trial had CRC as their primary cancer, of which just 9 received SABR. Despite these limitations, the survival benefit outweighs the risks and highlights SABR’s value as a therapeutic approach for long-term disease control in select patients. The SABR-5 trial showed similar findings, reporting excellent results, especially in regard to LC, with rates at the 1- and 3-year points being 93% and 87%, respectively, for patients treated with SABR [40]. The ORCHESTRA trial highlights the use of SABR alongside chemotherapy as an approach for managing extensive metastatic CRC [20]. By adding tumour debulking (including SABR, resection, or thermal ablation) to chemotherapy in patients with multiorgan metastasis, the authors showed that local treatment did not interfere with systemic therapy, as 89% of patients could resume chemotherapy after debulking. This trial therefore indicates that SABR does not preclude further systemic treatment and may provide additional disease stability. The phase II trial by Scorsetti et al., which exclusively evaluated inoperable colorectal liver metastases, achieved an LC rate of 91% at 2 years and a median OS of 29.2 months [41]. Notably, patients maintained an OS rate of 65% at 24 months, highlighting the potential of high-dose SABR (75 Gy in 3 fractions) to achieve durable disease control in colorectal liver metastases. Most impressively, the absence of grade 3 or higher toxicities further highlights SABR’s safety profile in a radical setting (Table 1).

SABR inflicts severe DNA damage in colorectal cancer (CRC) oligometastatic cells—particularly clustered DNA double-strand breaks [42]. The fate of cells undergoing SABR therapy depends on the effectiveness of the two double-stranded break repair pathways: non-homologous end joining (NHEJ) and homologous recombination (HR) [43]. CRC cells can survive radiation by rejoining broken DNA ends, but repair failure can lead to mitotic issues or apoptosis. TP53 is a tumour suppressor gene which is a crucial mediator of the DNA damage response, halting the cell cycle and inducing apoptosis when double-stranded breaks are irreparable. CRC cells with mutated TP53 show blunted apoptotic signalling and an impaired G1 checkpoint, allowing the further spread of radiation damage [44]. Defects in HR repair can increase radiosensitivity, as demonstrated in tumours containing BRCA1/2 mutations, which displayed improved tumour control after radiotherapy due to an inability to repair SABR-induced double-stranded breaks [43]. The DNA mismatch repair (MMR) system, though mainly responsible for correcting replication errors, also influences the SABR response. MMR-deficient CRCs are characterised by microsatellite instability and accumulate mutations which can lead to an altered DNA damage tolerance. Clinically, deficient MMR CRCs have shown lower pathologic complete response rates to neoadjuvant chemoradiation, suggesting an inherent radioresistance in MSI-high tumours [45]. The balance between SABR-induced DNA damage and the tumour’s DNA repair capacity—characterised by pathways like NHEJ, HR, and MMR—is a key factor to consider if CRC oligometastasis is to be treated.

**Table 1 ijms-26-10302-t001:** Comprehensive summary of SABR trials.

Trial Name (Author)	Study Type	Control Arm	Primaries, Number of Patients, (Number of Colorectal Patients), and Lesion Location	Dosage Regimen	Progression-Free Survival	Overall Survival	Local Control	Toxicity Rates	Summary of Published Outcomes
Phase II prospective trial on SBRT for unresectable liver oligometastases from colorectal cancer (Scorsetti, 2015) [41]	Phase II prospective	Not reported	Inoperable colorectal liver metastases **(n = 42 CRC)**	A dose of 75 Gy in three consecutive fractions of 25 Gy	**Median PFS: 12 months**	**Median OS: 29.2 months** **1 yr: 85.2%** **3 yr: 31.3%** **5 yr: 18%**	**2 yr actuarial LC rate: 91%**	78% grade 2 toxicity. No grade 3+ toxicity, RILD, or bile duct stenosis observed.	SBRT is a feasible alternative to surgery in inoperable tumours with good overall survival (29.2 months) and local control (91%).
Phase II study of individualised SABR of liver mets (Hong, 2017) [46]	Phase II single-arm, single institution	Not reported	Liver metastases from solid tumours, n = 89 **(n = 34 CRC)**	30–50 Gy in five fractions (based on effective volume of liver irradiated)	Cohort median PFS: 3.7 months 1 yr: 24.7% 3 yr: 9.2%	Cohort median survival: 18.1 months 1 yr: 66.3% 2 yr: 35.9% 3 yr: 20.8%	**Colorectal-specific 1 year LC: 58.8%** **3 yr LC: 44.7%**	87.6% experienced radiation-related toxicity, most commonly fatigue (68.5%), dermatitis (47.2%), and abdominal pain (23.6%). No grade 3+ toxicity observed.	CRC primary tumours had lower LC rates, but PFS and OS were comparable. TP53 and KRAS associated with poor prognosis.
Phase I dose escalation study and phase II study on SBRT for CRC hepatic mets (McPartlin, 2017) [47]	Phase I and phase II	Not reported	Colorectal liver metastases **(n = 60 CRC)**	Prescription dose of 33 to 57 Gy in six fractions	**Median PFS: 10.8 months**	**Median OS: 16 months** **1 yr: 63%** **2 yr: 26%** **4 yr: 9%**	**1 yr: 50%** **2 yr: 32%** **4 yr: 26%**	Mostly well tolerated; one case of grade 3 nausea, two cases of grade 3 thrombocytopenia (one resolved, one fatal). No grade 3+ liver toxicity, RILD, or late gastrointestinal complications.	Treatment is safe and may be associated with long term cure. Local control is better with higher SBRT dose.
Multicentre phase II on safety and feasibility of SABR for patients with oligometastatic cancer (Sutera, 2018) [48]	Multicentre prospective phase II	None described	Oligometastatic cancer; lung, colorectal, head and neck, etc., n = 147 **(n = 31 CRC)** Multiple lesion locations including lung, lymph node, bone, and liver	Depended on lesion size and location	**Colorectal median (distant) PFS: 10.4 months** Cohort (local) 1 yr PFS: 91% Cohort (local) 5 yr PFS: 75%	**Colorectal median OS: 54.4 months**	“excellent”, no values reported	Acute grade ≥ 2: 7.5%, grade ≥ 3: 2.0%. Late grade ≥ 2: 1.4%, grade ≥ 3: 1.4%. Grade 4 small bowel obstruction (n = 1). No significant quality-of-life decline.	High overall survival rates and a respectable progression free survival, with “excellent” local control.
SABR-COMET (Palma, 2020) [18]	Phase II Randomised	Standard of care (palliative systemic therapy)	Oligometastatic cancer; breast, colorectal, lung, prostate, etc., n = 99 **(n = 18 CRC)** Multiple lesion locations including lung, bone, liver, and adrenal	Allowable doses ranged from 30 to 60 Gy in 3–8 fractions	Cohort Median PFS: 5.4 months vs. 11.6 months 5-year PFS: 0% vs. 17.3%	Cohort Median OS: 28 months vs. 50 months 5-year OS: 17.7% vs. 42.3%	Cohort overall long-term LC rate: 46% vs. 63%	Grade ≥ 2 adverse events occurred in 29% (19/66) of the SABR arm vs. 9% (3/33) in the control arm (*p* = 0.03). Treatment-related deaths occurred in 4.5% (3/66) of SABR patients.	SABR improved overall survival by median 22 months.
ORCHESTRA trial (Gootjes, 2020) [20]	Phase II randomised	Chemotherapy standard of care	Oligometastatic colorectal cancer **(n = 88 CRC)** Multiple lesion locations including liver, lung, and lymph node	Tumour debulking, using SABR, resection, or thermal ablation, was added alongside chemotherapy; six radiotherapy sessions delivered	Not explicitly reported, focus on chemotherapy compatibility	Not specified (data collection still underway)	High, but variable across metastatic sites	SAEs in 50% of patients; grade ≥ 3 surgical complications in 24%, plus one possible SABR-related death from pneumonitis.	Local interventions like SABR did not interfere with chemotherapy. Stable disease control reported.
SABR-5 trial (Baker, 2022) [40]	Single arm phase II	None described	Oligometastatic cancer; prostate, colorectal, breast, lung, renal cell carcinoma, n = 381 **(n = 63 CRC)** Multiple lesion locations including bone, lung, lymph node, liver, and adrenal	24–60 Gy in 2–8 fractions	Cohort median PFS: 15 months 1 yr PFS: 56% 3 yr PFS: 31%	Cohort median OS: not reached 3 yr OS: 71%	Cohort 1 yr LC: 93% 3 yr: 87%	Grade 2+ toxicity: 18.6%, grade 3+: 4.2%, grade 4: 0%, grade 5: 0.3%. One possible SABR-related death due to biliary stenosis and infection.	Low rates of local failure, high median PFS, no specific colorectal data.
SABR-COMET-3	Phase III randomised (ongoing)	Standard of care (results pending)		*Pending results*	*Pending results*	*Pending results*	*Pending results*		*Pending results*
SABR-COMET-10	Phase III randomised (ongoing)	Standard of care (results pending)		*Pending results*	*Pending results*	*Pending results*	*Pending results*		*Pending results*
Alliance (A022101/NRG-GI009)	Phase III randomised (ongoing)	Systemic therapy alone (results pending)		*Pending results*	*Pending results*	*Pending results*	*Pending results*		*Pending results*

**Colorectal cancer-specific data is in bold.** Abbreviations: CRC, colorectal cancer; OS, overall survival; PFS, progression-free survival; LC, local control; SBRT, stereotactic body radiation therapy; SABR, stereotactic ablative radiotherapy; Gy, gray (unit of radiation dose); mets, metastases; RILD, radiation-induced liver disease; SAE, serious adverse events.

The literature suggests that there is a clear dose–response relationship with LC and OS, as evidenced by Petrelli et al. in their systematic review which analysed 656 patients with colorectal cancer liver metastases. The metanalysis found that there was a pooled 1- and 2-year survival rate of 67% and 56%, with a local control rate of 67% and 59%, respectively. They also found a significant linear relationship between the BED10 dose and OS and LC at the 2-year point. Based on correlation analysis, the study found that, for every increase of 1 Gy in BED10, the LC and the OS would increase by 0.21 and 0.11% [49]. This is further corroborated by Dell-Acqua et al., who found that a BED10 greater than 75 Gy was associated with a greater LC on multivariate analysis. This single-arm study concluded that the SABR prescribed dose depends on lesion volume and location [50].

CRC metastases tend to be radioresistant, as reported by Blinkley et al., with a local failure rate of 42.2% compared to 9.9% in other histologies, underscoring the need for a higher BED10 [22]. This is similarly reported by Jingu et al. in their systematic review of 18 studies which focused on pulmonary oligometastases from colorectal cancer. The study concluded that LC was significantly worse for colorectal pulmonary oligometastases compared to pulmonary oligometastases from other cancers, with a good LC only achieved through dose escalation [51]. Subanalysis of the SABR-5 trial also reflects this, with the authors noting that colorectal tumour histology was associated with a higher rate of local recurrence. Cao et al. suggest that there is an inherent radioresistance with CRC oligometastases that requires a higher BED10 to achieve a satisfactory LC [52]. These findings suggest the need for a higher-than-average BED10 to effectively treat CRC oligometastases.

Internationally, the European Society for Medical Oncology (ESMO) recommends SABR as a safe alternative treatment for oligometastatic CRC in patients not fit for surgery or other ablative treatments (Figure 1). The guidelines further highlight SABR as a treatment option in cases of unresectable liver metastases [53]. The American Society for Radiation Oncology (ASTRO) does not have colorectal-specific guidelines, but they have developed general SABR guidelines following a review of evidence by an expert panel [54]. These guidelines outline SABR as a feasible treatment option for recurrent or metastatic tumours that cannot be irradiated effectively or resected. In the UK, the latest inclusion criteria for SABR treatment in CRC oligometastases of the liver include, but are not limited to, 1–3 metastatic sites with no site exceeding 6 cm, unresectable metastases, predicted life expectancy of more than 6 months, adequate organ function, and Class A Child–Pugh score [17]. There is further guidance on the recommended dose-fractionation schedules that can be used for the treatment of CRC oligometastases. The suggested regimens include: 40–60 Gy in 3 fractions, 50–60 Gy in 5 fractions, and 30–60 Gy in 10 fractions.

It can be difficult to compare SABR as a treatment to surgery, as SABR is generally indicated for unresectable metastases in more advanced disease settings, so the patients are likely to have a poorer baseline. Min Lee et al. retrospectively compare SABR to wedge resections in the treatment of CRC pulmonary oligometastases [55]. The study found that there were no significant differences in the 3-year LC rate, with surgery achieving an LC of 88.6% compared to the 86.7% achieved by SABR. Another study also examined dose escalation in SABR for pulmonary and liver CRC oligometastases and found that higher radical doses can offer LC rates that approach surgery [16]. These studies highlight the efficacy of SABR as a treatment which rivals surgery in terms of outcomes, without the morbidity and recovery time associated with surgery.

SABR still has many limitations in the treatment of CRC oligometastases. For example, there is a lack of phase III trials investigating SABR in oligometastases, with the SABR-COMET trial being one of the few landmark trials that compare SABR to a standard of care. However, even in the SABR-COMET trial, there were only 18 of 99 patients who had CRC oligometastases. Similarly, the SABR-5 trial examined various types of oligometastases, with only 63 of 381 patients with colorectal oligometastases. There is also an emerging role for immunotherapy in conjunction with SABR, as evidenced by the ORCHESTRA trial, which found this treatment strategy to be feasible in cases of metastatic colorectal cancer [20]. Optimising the delivery of SABR to ensure maximum accuracy and reduced radiation exposure to OAR is another consideration in this field. These are all areas which warrant further investigation.

### 2.2. Advancements in Patient Selection and Prognosis

Stereotactic ablative radiotherapy has demonstrated a significant efficacy in improving disease outcomes in patients with oligometastatic CRC and has thus emerged as a promising treatment modality. However, it is unfortunately the case that currently only a subset of patients treated with SABR achieve long-term survival, with a median 5-year survival ranging between 30 and 50% [18]. There are various genomic, pathological, anatomical, and clinical factors that have been identified as critical determinants of outcomes in patients with oligometastatic CRC that are undergoing SABR therapy, and a deeper understanding of these factors may allow for a more effective patient selection in current SABR protocols (Figure 2).

#### 2.2.1. Anatomical Factors

Various reports have demonstrated that the site of metastasis plays a crucial role in predicting treatment outcomes in CRC patients receiving SABR. The two most common sites of metastasis in CRC are the liver and the lungs, with the metastases in the latter being associated with superior local control rates. Ahmed et al. [23] examined 29 lesions (12 lung and 17 liver) from an independent cohort of 23 patients who were treated for oligometastatic CRC. They found that the 24-month local control rates were 100% for lung metastases compared to 73% for liver metastases, suggesting that the anatomical site of metastasis can play a significant role in determining the efficacy of SABR therapy [23]. Using a de-identified meta-data pool, the authors demonstrated that liver metastases have a higher radioresistance than lung metastases (with Radiosensitivity Index (RSI) values of 0.43 and 0.32, respectively), suggesting the difference in local control rates may be attributable to differences in the radioresistance of metastatic lesions in the two anatomical locations, although RSI values for the 29 lesions were not directly collected. Similarly, patients with bone lesions have been shown to have poorer local control rates and decreased overall survival compared to those with metastases at other locations [56].

The variation in radiosensitivity between anatomical sites of metastasis is likely a reflection of the differences in parenchymal histology and associated vasculature, which in turn will give rise to differences in the tumour microenvironment [23]. In particular, there are several features of the liver that make it intrinsically susceptible to metastases, including the slow and tortuous flow through a dual circulation, as well as the presence of liver–sinusoidal endothelial cells, which are rich in surface molecules and have fenestrations which facilitate the attachment and enhanced access of circulating tumour cells [57]. Furthermore, the liver exhibits regional immune suppression due to its constant exposure to potentially inflammatory stimuli, which gives rise to a relatively tolerant tumour microenvironment. This is at least in part mediated by hepatic stellate cells, which have been shown to be potent inhibitors of T cell responses, mediated by the inducible expression of B7-H1, an inhibitor of the B7 family [58]. Considering that a key component underlying the efficacy of radiotherapy is the enhanced activation of the anti-tumour immune response, the relatively cold immune landscape of the liver may partly explain the poor radiosensitivity of liver metastases compared to metastases in tissues with a more active immune landscape [59].

Alternatively, it may also be the case that more radioresistant primaries have an increased propensity to metastasise to the liver. Indeed, it has been demonstrated in preclinical models that radiotherapy-resistant colorectal cancer cells have features that facilitate liver metastasis. Using RNA sequencing, these cells were shown to have upregulated the protein RSG14, which competitively inhibited the GSK-3β-mediated phosphorylation of β-catenin, subsequently leading to nuclear translocation and downstream transcriptional changes to facilitate tumour epithelial–mesenchymal transition (EMT) [60]. Furthermore, RSG14 upregulation was also associated with the formation of a pre-metastatic niche that facilitated liver metastasis, including the pro-tumorigenic polarisation of tumour-associated macrophages and decreased CD8+ T cell activation. The mutational status of primary tumours has also been related to the site of metastasis, with primary tumours harbouring KRAS mutations more commonly metastasising to the lung compared to the liver [61]. This is particularly interesting considering KRAS mutations are associated with poorer outcomes in SBRT [56], therefore suggesting that molecular features must be considered within the anatomical context when predicting the efficacy of SBRT.

Alongside the site of metastases, the tissue of origin also plays an important role in determining the response to therapy. In the case of colorectal cancer, this refers to whether the primary cancer originates from the colon or the rectum. It has been found that lesions derived from primary rectal tumours exhibit poorer outcomes when treated with SABR compared to lesions originating from primary colon tumours. A study of 87 lesions from 53 patients treated with SABR found that lesions derived from colon tumours had a 2-year local control rate of 87.4% compared to 55.1% in metastases derived from rectal tumours (HR. 4.7; *p* = 0.001), suggesting that lesions derived from the rectal tumours are associated with a higher rate of relapse [21]. Furthermore, when comparing lesions treated with the same radiation dose (60 Gy in three fractions), they showed that the primary site continued to predict local relapse, suggesting a difference in radiosensitivity between metastases derived from rectal vs. colon primaries. A similar pattern was observed by De Baere et al. (2015), who observed a 15% decline in local control following the treatment of lung metastases with radiofrequency ablation if the primary tumour was rectal compared to colonic [62]. The difference in radioresistance between lesions originating from rectal vs. colon primaries is likely related to differences in their mutational profiles. Indeed, rectal primaries have been shown to possess higher rates of KRAS mutations [21] and TP53 overexpression [63] compared to colonic primaries, which may explain the improved local control in the latter. Overall, it would be beneficial to tailor treatment strategies to the primary histology and anatomical context of the patient’s metastases, with different dosages or adjunctive therapies offered to patients with worse-performing metastases.

#### 2.2.2. Genomics

Genomic alterations have also been implicated as important determinants of response to SABR, with several key mutations being associated with treatment failure and poorer survival outcomes. Interestingly, recent work by Wang et al. has highlighted the distinct treatment sensitivity of rectal cancers with deficient mismatch repair/microsatellite instability-high (dMMR/MSI-H) genotypes [64]. In a multicentre propensity score-adjusted analysis of 119 patients with locally advanced rectal cancer, they found that surgery alone was associated with better OS and PFS compared to surgery + chemoradiotherapy, suggesting that certain genotypes may confer an intrinsic resistance to conventional cytotoxic and radiotherapy-based regimens. Jethwa et al. conducted an analysis of 109 lesions across 85 patients who received SABR for oligometastatic CRC with the aim of identifying genomic factors associated with treatment efficacy and survival [56]. They found that KRAS mutations and combined KRAS and TP53 mutations were associated with a worse overall survival. The latter was also associated with a higher risk of local failure (LF), with a 44% 1-year cumulative incidence of LF compared with 11% for patients without combined *KRAS* and *TP53* mutation. Interestingly, TP53 mutations alone did not produce a significant reduction in overall survival or local control. Similarly, BRAF mutations were not associated with poorer survival. In contrast, Gui et al. found that TP53 driver alterations were significantly associated with a higher risk of intracranial progression following stereotactic radiosurgery for CRC brain metastases [65]. Another study showed that the presence of a KRAS mutation in codons 12, 13, or 61 was the strongest predictor of poor local treatment outcome, with 1-year local control rates of 42.9% compared to 72.1% for tumours without detected mutations [46]. Furthermore, patients with both KRAS and TP53 mutations experienced the poorest treatment outcomes, with 1-year local control rates of 20%. However, only 34 of the 89 primary tumours analysed in this study were CRC. Out of the CRC tumours, three had the double mutant KRAS and TP53 genotype, all of which exhibited local failure.

Clearly, KRAS mutations are a strong predictor for the poor local control of oligometastases treated with SABR (Table 2). Therefore, determining the tumour genotype prior to commencing treatment may allow for a more accurate prognostication and patient stratification, with the possibility to intensify treatment in particular patient subgroups. However, KRAS-mutated cancers are still relatively heterogenous, and thus additional biomarkers are necessary to accurately predict treatment responses. For example, the combination of mutations in KRAS and TP53 defines a subset of extremely radioresistant tumours, which are likely to require a more intensive approach to therapy, likely with higher doses or adjunctive treatments to help improve responses. The mechanisms underlying the enhanced radioresistance in KRAS-mutated tumours have been studied using colorectal cancer cell lines, which revealed that mutated KRAS isoforms activate EGFR and H-Ras, resulting in enhanced cellular radioresistance through PI3K/AKT signalling [66]. Moving forward, gaining a mechanistic understanding of how concomitant TP53 mutation modifies the radiosensitivity of KRAS-mutated tumours may allow for the development of more targeted therapies. One suggestion has been to “radiosensitise” such tumours, with PKC, Aurora B, or Chk1 emerging as molecular targets for radiosensitisation that warrant further research going forward [67,68]. A promising radiosensitiser is L-19-IL2, which is an immunocytokine that has been shown in preclinical models to deliver IL-2 to tumour cells via the selective L-19 dependent binding of the extra domain B of fibronectin located on tumour vasculature endothelium [69]. Van Limbergen et al. (2021) conducted a phase I trial in six patients, demonstrating the safety of using L-19-IL-2 at its recommended dose (15 million IUs), reporting a subset of long-term progression-free responders [70]. This study then led to the ongoing phase II trial ImunnoSABR [71]. Further work must characterise whether there is a clinical benefit to such radiosensitisers and must carefully consider whether any observed clinical benefits can be attributed to radiosensitisation or alternatively arise from unrelated abscopal effects.

#### 2.2.3. Clinical Factors

The time to treatment and tumour volume have also been considered as clinical factors related to treatment efficacy. A delay in initial treatment has been associated with more advanced disease that requires several lines of treatment, leading to a mutational shift that may cause radioresistance [21]. However, there has been conflicting evidence as to whether the number of systemic treatments administered before SABR therapy correlates with local control rates, with some studies showing no association and others suggesting that previous therapy is associated with a significantly lower LC [73]. Alternatively, local control has been shown to be superior in smaller-volume tumours, with one trial demonstrating a 100% 2-year local control rate in lesions with a maximum diameter of 3 cm. However, the relevance of this association for predicting treatment outcomes has been debated based on the fact that larger lesions are typically prescribed lower BED schedules [74].

#### 2.2.4. Biomarkers

Finally, beyond the consideration of anatomical, genomic, and clinical factors for predicting the response to treatment, it may also be useful to identify biomarkers for measuring treatment response and predicting recurrence. An analysis of post-treatment circulating tumour-specific DNA (ctDNA) has been suggested as a possible prognostic indicator for the local treatment of CRC. A pilot study using a highly sensitive ddPCR assay analysed the presence of mutation-specific ctDNA in the plasma samples of 35 patients following treatment of liver and/or lung metastases, and found that patients that have post-treatment blood samples positive for ctDNA demonstrated a shorter median time to recurrence [75]. Having such biomarkers available to measure treatment response could be effective for identifying patients who may require additional interventions. Recent findings by Wilkins et al. demonstrated that baseline immune gene expression profiling can stratify rectal tumours into radioresistant and radiosensitive phenotypes, with poor responders showing an upregulation of stromal and immunosuppressive genes such as CD274 (PD-L1), IL6R, and CD163, which were associated with inferior outcomes [76]. Importantly, their study also revealed that good responders exhibited a radiation-induced transition to an immunologically “hot” tumour microenvironment, characterised by the upregulation of interferon-gamma and T cell cytolytic gene signatures—highlighting the potential of longitudinal immune biomarkers in predicting responses to radiotherapy. Overall, it is evident that many interrelated factors play a part in determining the efficacy of SABR in the treatment of oligometastatic CRC, and an appreciation of these is required to ensure an optimal patient selection, stratification, and prognostication.

### 2.3. Optimising SABR

Optimising SABR delivery in oligometastatic colorectal cancer requires a careful consideration of disease site-specific challenges, as the location of metastases often dictates the technical complexities and risks involved. The most common sites of oligometastases in CRC are the liver, lungs, and pelvic or para-aortic lymph nodes—each presenting unique obstacles. For example, liver lesions and upper abdominal nodes are subject to significant respiratory motion, requiring motion management strategies such as gating, tracking, or abdominal compression. In contrast, recurrent pelvic nodal disease poses a higher risk of gastrointestinal toxicity, particularly to the bowels, which demands planning to protect adjacent organs at risk (OARs). When lesions are located near bony structures, such as the spine, techniques like bone tracking can be used to enhance accuracy. Therefore, the choice of radiotherapy platform and planning approach should be tailored to the anatomical site, ensuring the maximum therapeutic benefit while minimising treatment-related morbidity.

#### 2.3.1. Magnetic Resonance-Guided SABR

A promising advance to the treatment of colorectal oligometastases with stereotactic ablative radiotherapy is the use of magnetic resonance (MR) guidance to improve treatment accuracy through better soft tissue visualisation and the ability to finely delineate tumour borders. Typically, the planning target volume (PTV) includes a clinical target volume (CTV) with an added margin allowing for uncertainties that may arise from variables such as movement or poor resolution. MR-guided systems are well positioned to leverage greater spatiotemporal accuracy, thereby minimising the PTV, enabling lower doses to healthy tissue and, consequently, lower complication rates.

However, MR-guided systems are subject to unique causes of their own uncertainty. In MR-guided radiotherapy (MRIgRT), the dose optimisation must account for the magnetic field due to the electron return effect. Electrons are deflected by the magnetic field used and are thus returned to the skin’s surface, increasing the total surface dose. Whilst JJ Lagendijk et al. (2013) suggest that this return may be advantageous in the treatment of breast cancer, it is unlikely that surface return can be anything more than a hinderance to the treatment of colorectal cancer oligometastases [77]. Importantly, the previous literature has demonstrated this using GafChromic film in poly-methyl methacrylate (PMM) air phantoms which were irradiated on the Elekta Unity system. Shortall et al. (2020) validated Monte Carlo-predicted ERE perturbations around spherical air cavities, simulating bowel gas, highlighting the challenges of treating colorectal oligometastases where such cavities are common [78]. Clinically, this is highly relevant, as bowel gas is frequent in patients with colorectal liver metastases, meaning ERE could directly increase gastrointestinal toxicity—a critical determinant of treatment tolerance and quality of life. This experimental verification highlights that dosimetric distortions are not only theoretical but can have a real clinical impact. Such work challenges clinicians and researchers alike to find bowel preparation strategies or motion management protocols that might mitigate this toxicity.

Beyond pure dosimetric consequences, the electron return effect modifies the microdosimetric distribution of secondary electrons, leading to local increases in linear energy transfer (LET) at the skin surface. These LETs are biologically significant as they give rise to clustered DNA double-strand breaks that are more resistant to repair by canonical NHEJ pathways. Nevertheless, whilst this may enhance tumouricidal effects in certain superficial malignancies, in the setting of colorectal oligometastases treated in deep tissues, this altered profile predominantly increases epithelial toxicity without any notable offset in therapeutic benefit.

MR-Linac systems, which deliver a simultaneous imaging modality with radiation delivery, are also subjected to degradation by geometric distortions, creating uncertainty between imaged and real anatomy. Several factors coalesce, giving rise to geometric distortion. The most notable is gradient non-linearity (GNL), which arises from the practical limitations of manufacturing and engineering to create a magnetic field gradient that varies perfectly linearly across the entire image field. In reality, deviations from this ideal state lead to image distortions, which is problematic in the setting of SABR. Despite commercial scanners providing corrections to mitigate distortions, residual distortion remains—especially away from the scanner’s central imaging axis (off-axis regions). These residuals are problematic because MRI-Linacs rely heavily on the geometric accuracy of images for real-time beam adaptation techniques such as multi-leaf collimator (MLC) tracking. Distorted images misdirect the radiation beam, causing it to target incorrectly positioned anatomical structures. In CRC oligometastases, which are often close to the bowels and major vasculature, even millimetre-level mis-targeting risks underdosing tumour edges and overdosing critical organs, potentially reducing local control whilst increasing acute toxicity. Moreover, patient positioning constraints on MRI-Linacs often require treatment targets to be placed away from the imaging centre, increasing the susceptibility to distortion-related inaccuracies. Such issues are exacerbated when considering the relatively rigid and immobile nature of most MRI-Linac patient couches. However, an advantage of MR-Linac over systems like Cyberknife is that MR-Linac does not require fiducial insertion, which is invasive and not appropriate for all patients [79].

However, on-demand geometric adaptation based on tumour or at-risk organ anatomy has been well documented. Liu et al. 2023 described a protocol to develop and integrate a real-time distortion correction algorithm that enables accurate real-time adaptive radiotherapy through accurate MLC tracking [80]. The group imaged a phantom containing 3718 markers to quantify geometric distortion, with positions fitted utilising an 8th-order spherical harmonic model. This allowed the creation of a distortion vector field representing the pixel-level distortion across the image planes. This vector field was calculated and then used to rapidly (<10 ms per frame) correct real-time cineMRI images. Prior to correction, Root-Mean-Square error ranged from 1.2 mm up to 3.3 mm, which was reduced after correction to approximately 0.8–1.2 mm. Additionally, the maximum tracking error was reduced as well from as high as 6.4 mm (uncorrected) to 2.7 mm (corrected). Furthermore, such corrections only minimally affected latency (additional latency < 16 ms), maintaining an overall system latency within acceptable clinical limits (335–545 ms). Whilst such work effectively demonstrates the feasibility and benefits of integrating real-time distortion correction in MRI-guided radiotherapy, limitations such as its current confinement to the 2D plane correction and reliance on simpler image-domain corrections highlight areas for further development. Future research must focus on implementing advanced 3D distortion correction techniques, potentially incorporating emerging technologies such as machine learning-enhanced k-space reconstruction methods to comprehensively address these limitations and exploit the benefits of such an approach.

Indeed, the success of MRgSBRT in the face of MR-specific uncertainties is highlighted by Rosenberg et al. 2019 [81]. Using real-time sagittal cineMRI, tumours or surrogate anatomy were tracked in order to use small margin treatment (2–5 mm) individualised based on patient anatomy and geometric distortions. A total of 26 patients with primary liver cancers or metastatic liver tumours were treated between 2014 and 2017. Regiments included a median dose of 50 Gy in five fractions limited to a mean liver dose of <15 Gy, and patients were assessed every 3–6 months for local progression (LP), OS, and toxicity, graded according to the NCI criteria. The freedom from local–regional progression (FFLP) at a median follow-up of (21.2) months was 80.4%, and OS was 69% at 1 year and 60% at 2 years. Importantly, researchers demonstrated the safety of MRgSBRT, noting a minimal-grade ≥ 3 gastrointestinal toxicity (7.7%) and no grade 4 or 5 toxicities. Furthermore, only two patients showed a decrease in liver function, and both had high tumour volumes or high liver doses from prior treatments. Ablative regimens are thought to trigger immunogenic cell death (ICD) in addition to providing local control. ICD is characterised by calreticulin exposure, HMGB1 release, and activation of the cGAS-STING pathway, which together prime anti-tumour immunity [82]. It has been suggested that MR-guided prevision allows a sufficient dose per fraction to consistently induce ICD whilst limiting collateral lymphocyte depletion. It follows that such an approach is favourable for synergistic therapy with immune checkpoint inhibitors, which reinvigorate pre-existing anti-tumour immunity. Nevertheless, whilst Rosenberg et al. provide compelling early evidence supporting MR-guided SBRT for liver tumours, applications to oligometastatic colorectal cancer remain uncertain. Colorectal metastases had a slightly lower control (75%) than primary tumours (100% for HCC) or other metastatic lesions (83%). This disparity underlies the importance of MR-guided adaptive planning in CRC specifically, as daily re-planning and margin reduction may help overcome the lower baseline tumour control compared to other histologies. Such a result emphasises the potential of daily adaptive MR-guided treatments to escalate dose safely near critical organs, which is particularly important in the treatment of colorectal liver metastases. Future clinical trials aim to explore the combination of MRgSBRT and adaptive planning with dose escalation, which may provide more promising results in the challenging setting of treatment for colorectal metastases.

Functional imaging modalities such as MR perfusion and oxygenation mapping may represent an additional opportunity afforded by MR-guided platforms. Hypoxic tumour subvolumes, characterised by HIF-1a stabilisation and the upregulation of angiogenic factors like VEGF, are well-recognised mediators of radioresistance [83]. Tumour hypoxia being one of the strongest determinants of radioresistance primarily relates to the “oxygen effect”. In the presence of molecular oxygen, at or instantaneously around the time of onset, low-LET radiation ionises water molecules to produce high-energy electrons and reactive oxygen species (ROS), which cause indirect DNA damage. It follows that, in anoxic conditions, ROS formation is abrogated, reducing DNA damage at a given dose. Moreover, much work suggests that the dose required to achieve an equivalent dose kill in anoxic versus oxic conditions—known as the oxygen enhancement ratio (OER)—is approximately 2.5–3 for low-LET photons. OER modelling further implies a non-linear relationship, with pO2 highlighting a significant radioresistance when pO2 falls below 10–15 mmHg. Dose-painting approaches guided by MR perfusion may enable the selective escalation to hypoxic niches to mitigate resistance at the molecular level [84]. Recent systematic evidence confirms that MRI-derived biomarkers can correlate with hypoxia-related factors, which might be integrated into workflows of the future [85]. In this way, MR-guided SABR holds the promise of unifying precise geometric targeting with molecularly informed radiosensitisation strategies.

When treating oligometastatic colorectal cancer with SBRT, another challenge facing clinicians is patient breathing and baseline shifts and drifts which cause target movement. Small adjustments lead to potential tumour underdosage or surrounding healthy tissue overdosing, which is currently inadequately managed by surrogate markers and margins. Tumour trailing is a technique that continuously adapts the radiation beam position based on recent tumour location data in order to compensate for this motion and improve MRI-guided SBRT. Fast et al. (2019) simulated the treatment of 17 patients with oligometastatic liver disease via three fractions of 20 Gy and intensity-modulated radiotherapy (IMRT), planned using mid-position CT scans [86]. A phantom study was performed to confirm the simulation results to treatment where the beam aperture was varied continuously according to the tumours’ average position from the past three breathing cycles. Notably, tumour trailing consistently improved tumour coverage by reducing the geometric miss caused by baseline movement. The median tumour dose improved notably in all modelled scenarios. In realistic drifts combining linear, single, and periodic shifts, a median increase in the range of 0.5–15 Gy was noted. Furthermore, the Gamma pass rate nearly doubled with tumour trailing (49% to 98% at 3%/2 mm), indicating a significantly improved radiation targeting accuracy. Such a modality seems highly promising for the treatment of colorectal cancer given it does not require complex prediction algorithms or very high frequency imaging, maintaining a practical feasibility and potential for integration into other correction techniques. Nevertheless, such findings require extensive testing in clinical settings focusing on oligometastatic colorectal cancers. Previous work has suggested that liver movement is largely confined to the superior, inferior, and anteroposterior directions, which simplifies trailing and may not be the case in more complex colorectal oligometastases. Furthermore, the group examined only single lesions due to the difficulty in managing the motion of multiple lesions simultaneously. Ensuring that there is homogenous intratumoural coverage is vital not only from a dosimetric perspective but also at the molecular level, where geographic miss may allow the persistence of radioresistent subclones with an intact DNA repair capacity. For oligometastases, where radioresistent subclones and clonal escape drive local progression, motion management strategies like tumour trailing are not only a technical advancement but may translate into meaningful improvements in progression-free survival. These misses are thought to mediate clonal escape and local progression that may translate, highlighting the importance of improving dose conformity.

#### 2.3.2. Proton-Based SABR

Other advanced delivery systems, such as proton-based SABR, have the potential to offer a significant advantage in treating liver metastases from colorectal cancer by leveraging the dosimetric precision of protons. The use of charged particles in lieu of high-energy photons relates to the nature of photons to deposit their energy along their beam path, thus depositing dosage beyond the site of the tumour. This exit dose leads to a larger volume of target irradiation, which is contrasted by protons that deposit energy at a defined depth (Bragg peak phenomenon) and therefore lack a significant exit dose. Hong et al. (2017) used an individualised Veff dosing strategy to allow for the safe delivery of high doses of radiation to liver metastases [46]. Importantly, in the setting of large metastases of 6 cm or greater, the group found proton-beam therapy to be effective, with a one-year LC rate of 73.9% and no grade 3–5 toxicities or cases of radiation-induced liver disease. Such a finding is particularly important when realising that photon-based SABR often excludes such large lesions. Nevertheless, despite the promise of showing overall feasibility, LC in CRC metastases was disappointingly low at only 58.8% compared to 72.2% in other adenocarcinomas and 94.7% in all other histologies (19 cases). Interestingly, a subgroup analysis of tumour genotypes revealed that CRC metastases containing KRAS mutations, especially in conjunction with TP53 mutations, were strong predictors of poor control. This again supports the notion that such tumours are largely radioresistant to both proton- and photon-based therapy, highlighting that this appears to be modality-independent and instead biology-driven.

Previous work in the literature suggests that higher biologically effective doses (BEDs > 100 Gy) correlate with better local control. In fact, this is a key limitation of the photon-based therapy, which is limited by toxicity due to a less refined ability to spatially control the dose, particularly in more mobile anatomical areas such as the bowels. Hong and colleagues largely used doses in the 30–50 Gy range, with only a single patient under a dosing schedule greater than or equal to 100 Gy. Accordingly, further work must evaluate the effectiveness of proton-beam therapy at treating CRC metastases with minimal toxicity using BEDs greater than 100 Gy. Such work may provide a therapeutic solution that can challenge tumours with radioresistant tumour types. Other limitations of proton-based SABR also involve higher costs, limited availability, and the requirement of specialised expertise, all of which restrict widespread clinical implementation [37]. Careful patient selection and improved proton therapy delivery methods are crucial to mitigate toxicity risks and improve clinical outcomes.

### 2.4. Synergy with SABR

#### 2.4.1. Immunotherapy

Combining SABR with immunotherapeutic modalities is an area of active investigation. There are some examples of case studies and small-scale clinical trials which have used immunotherapy in combination with SABR, and these have mostly shown limited clinical benefits, apart from in small subsets of patients. Given that immunotherapy is a growing field, there is still a significant scope for further investigation to determine whether these therapeutic modalities can synergize to achieve better outcomes. This section will discuss some of these studies, alluding to the abscopal effect of radiotherapy.

Microsatellite stable colorectal cancer shows resistance to immune checkpoint inhibition, and the disease has a poor prognosis in the oligometastatic state [87,88]. However, when in combination with radiotherapy, tumour cells become more susceptible to immunotherapy and immune killing, due to alterations to the tumour microenvironment without increasing systemic toxicity [87,89]. The larger fractionated doses delivered via SABR, as opposed to conventional radiotherapy, can cause significant changes to an anti-immune tumour microenvironment, leading to increased immune responses and increased susceptibility to immunotherapy. The proof of concept of the synergistic effect between radiotherapy irradiation and checkpoint inhibition has been demonstrated in both case reports and pre-clinical mouse models [88,90]

The literature supporting combination immunotherapy and SABR in colorectal cancer is not extensive. However, there are small-scale clinical trials which have attempted to demonstrate a proof of concept. The first example of SBRT–immunotherapy combination therapy was a pilot study of AMP-224, a fusion protein which binds to PD-1 T cells expressing high levels of PD-1 (exhausted T cells), but not to T cells expressing low levels of PD-1. A total of 17 patients received SBRT at 8 Gy in one or three daily fractions, then were treated with 10 mg/kg of AMP-224 weekly; however, no objective responses were observed in the colorectal cancer population [91]. Additional studies tested AMP-224 alongside both SBRT and cyclophosphamide; however, once again, no objective response was noted, and overall survival rates were just 6 months [92]. In this study, there was a 20% cancer control rate, and there was a reversal in the polarisation of tumour-associated macrophages from a tumour-promoting M2 to an uncommitted M0 subtype. This suggests that there was an impact on the tumour microenvironment; however, this did not translate to a clinical benefit in 80% of cases.

Immune checkpoint blockade in isolation has poor results in pMMR advanced colorectal cancer, with a progression-free survival of just 2.2 months and overall survival of 5 months in the literature [93]. However, there is an increasing body of clinical evidence, particularly within the scope of non-small cell lung cancer, which suggests SBRT can enhance anti-cancer responses to ICB [94,95]. An international single-arm phase II study assessed this interaction within a colorectal cancer cohort using a dose of 45 Gy SBRT in three fractions (BED10 > 100 Gy) and three weekly atezolizumab IV infusions (1200 mg) [96]. The median overall survival was 8.4 months, and progression-free survival was just 1.4 months; however, there was a group of “elite responders” who demonstrated robust responses and had a median progression-free survival of 19.2 months. This suggests that there is a specific patient subset who derives benefits from SBRT and atezolizumab combination therapy with both pMMR and MSI-H colorectal cancers. RNA sequencing data showed a significant upregulation of genes known to increase T and B cell trafficking to tumours (CCL19, CXCL9) and tumour cell killing (GZMB) in these “elite responders”. The interaction between SBRT and the immune checkpoint blockade appears to have created an anti-tumour environment in these elite controllers. Further work is needed to understand why this response was only seen in this patient subset. This could set the stage for a more personalised approach to therapy, with patients that have characteristics which make them more likely to be “elite responders” being selected for a combination therapy with SBRT and atezolizumab. However, the current evidence suggests that the efficacy of a SABR–immunotherapy combination is limited in the general patient population.

On a molecular level, the mechanism of tumour cell death by SABR is multifaceted, incorporating double-strand DNA breaks, vascular damage mediating ischemic-induced cell death, and anti-tumour immune responses [97,98]. Double-strand DNA breaks are the primary mechanism at lower doses, and, as the dose increases, so do the tumour cell deaths secondary to vascular insufficiency and hypoxia. Immunologically, SABR acts as an in situ cancer vaccine by promoting the release of a substantial number of tumour antigens, leading to the targeted activation of the immune system.

The radiation from SABR induces tumour cell death, leading to the release of damage-associated molecular patterns (DAMPs), such as ATP, and tumour-associated antigens (TAAs) [99]. DAMPs such as ATP attract dendritic cells and promote their maturation, while cytosolic DNA released from dying tumour cells activates the cGas-STING-type I interferon signalling pathway in dendritic cells and stromal cells [97]. This signalling drives local inflammation, recruiting additional dendritic cells, and primes cytotoxic T cells. TAAs are then cross-presented by dendritic cells to CD8+ cytotoxic T cells within the tumour microenvironment, leading to a robust anti-tumour immune response [99]. However, the magnitude of this response depends on the immunogenicity of the DAMPs and TAAs and is dose-dependent [100]. Additionally, at high radiation doses, Trex1 exonuclease is upregulated in a dose-dependent manner, degrading cytosolic dsDNA and reducing STING activation, which limits the overall effectiveness of SABR [101]. Therefore, finding the correct SABR dosage regimen to maximise its effectiveness, while maintaining a robust immune function, is essential.

Immunotherapy can accentuate the effect created by SABR alone via checkpoint inhibition (PD-1 and CTLA4) or targeting other immune pathways such as GITR (glucocorticoid-induced tumour necrosis factor-related protein acts as a co-stimulatory receptor), promoting anti-tumour responses [102]. For example, a recent phase I/II study in the context of advanced solid malignancies (including colorectal cancer) combined BMS-986156, an agonist of GITR, with SABR and ipilimumab or nivolumab [100]. The treatment regimen was well tolerated, achieving a disease control rate of 48.7% and an abscopal effect rate of 46.2% in the cohort of advanced cancers. Here, disease control is mediated by systemic T cell activation, initiated by the release of DAMPs and amplified by co-stimulation with checkpoint inhibition.

Both safety and toxicity are key considerations when evaluating combination therapy, given that radiotherapy and immunotherapy both have notable side effects associated with them. There are a small number of examples in the literature of previous investigations of combination therapy in various metastatic solid tumours, including a collection of phase I trials with 123 patients with primary lung cancer or lung metastases (12% with colorectal primaries) [103]. Local control was deemed to be excellent in this study, with a 3.6% local failure rate at one year. Additionally, there was a low level of complications, with an 8.1% rate of grade 3+ pneumonitis from the combined therapy. Other phase I studies have supported these findings, with the combination of SBRT and pembrolizumab showing no signs of increased immunotherapy-related toxicity in a small cohort of 15 patients [104]. Given the body of evidence highlighting the potential of combination therapy, several trials are now recruiting to analyse the clinical benefits of combining SABR and immunotherapy for oligometastatic patients (Table 3).

#### 2.4.2. The Abscopal Effect

The abscopal effect of radiotherapy describes how ionising radiation at one tumour site can reduce tumour growth outside of the irradiated field at distant sites in metastatic cancer [105]. It has been described in preclinical and clinical studies and shown to be both immune-mediated and tumour-specific [106]. However, this effect is very rare in the published literature. Between 1969 and 2014, with a median radiation dose of 31 Gy, there were only 46 case reports of the abscopal effect in the literature [107].

The immunotherapy era has breathed new life into the hopes of using the abscopal effect to improve cancer treatment. As mentioned, in the pre-immunotherapy era, 47 cases of the abscopal effect were described; however, 47 cases treated with combination therapy (immunotherapy + radiotherapy) were reported between 2012 and 2018, suggesting that immunotherapy could revive the impact of the abscopal effect [74,106]. Mechanistically, cytosolic DNA from irradiated tumour cells activated the cGAS/STING pathway in dendritic cells, leading to type I interferon production [97]. In response to interferon signalling, tumours upregulate PD-L1 as a response to dampen cytotoxic T cell capabilities, limiting the systemic abscopal effect [108]. Checkpoint inhibitors, such as anti-PD1 and anti-CTLA4, block the upregulated inhibitory signals, sustaining T cell proliferation, cytokine production, and anti-tumour activity systemically [109]. This allows for activated tumour-specific CD8+ T cells to circulate in the bloodstream, recognising TAAs on distant metastases and eliciting a cell-mediated immune response at sites away from the site of irradiation. This is the abscopal effect [110]. A systematic review showed that dosage regimens for achieving the abscopal effect were varied, with a median BED10 of 49.65 Gy (range 28–151 Gy). Most patients received either ipilimumab, pembrolizumab, or nivolumab. In patients who experienced the abscopal effect, responses were robust, with complete or partial responses in 92% of the 24-patient cohort, and stable disease in the 2 patients without response [106].

The recent reports of instances of the abscopal effect may be a result of improving radiotherapy technology, such as the increasing use of SABR. SABR allows for larger doses of radiation to be delivered in a targeted fashion as compared to conventional radiotherapy, which likely increases the chance of systemic responses, as seen in the abscopal effect [111]. The phenomenon is not well understood, and the mechanisms behind it are thought to be the release of neoantigens due to radiotherapy injury leading to systemic tumour-specific immune response, resulting in the reduction in tumour growth at distant sites via the action of CD8+ T cells [74,112]. However, the small number of cases suggests that the conditions to observe the abscopal effect must be specific, and the ceiling of immune activation to observe systemic impacts must be high [106]. Overcoming the immunosuppressive tumour microenvironment is also a key challenge; immunotherapy will play a key role in addressing this challenge [112]. One line of research suggests that, given the relatively large increase in immune activation required, the abscopal effect might be maximised under high dose regimens. Two case reports that describe carbon-ion radiation in combination with immune checkpoint blockade appear to support this notion. Of note, Yamada et al. reported a 75-year-old man with recurrent colorectal cancer (T4N2M1 stage IV) with a mass in the right common iliac artery, which was not amenable to treatment given its proximity to the small bowel, that was reduced considerably and did not recur following a 73.6 Gy regimen delivered in 16 fractions over 28 days. It is, however, of note that the abscopal effect is poorly characterised in the literature. Such anecdotal evidence serves as an excellent proof of concept to suggest that off-“primary” tumour, on-target effects might be an unexpected benefit of increased dosing schedules. The clinicians of the future must endeavour to examine how systemic immune markers are modulated by different SABR profiles, but must take caution as to not invoke dangerous immunotoxic side-effects.

Overall, the abscopal effects of combined radiotherapy and immunotherapy could be a key player in the future of metastatic cancer treatment, allowing irradiation at one site to help treat systemic disease.

#### 2.4.3. AI and Machine Learning

AI and machine learning have a future in the treatment of diverse cancers using SABR; however, currently, applications are limited. Within SABR, one case study in a colorectal cancer patient used artificial intelligence tools to develop an intensity-modulated radiotherapy plan, maximising dosage to tumours while minimising exposure to surrounding tissues, thus demonstrating the future potential for incorporating AI into standard treatment, saving time and resources compared to generating these plans manually [113].

Similarly, Li et al. developed a CT radiomics-based AI model combining tumour imaging with clinical biomarkers such as CEA, CA19–9, age, and sex. The authors noted a development with high accuracy (AUC~0.90) in under five minutes. Such a result is significant, providing an excellent proof of concept for the development of a clinical screening tool to inform patients likely to develop liver metastases in a timely manner compatible with the pace of clinical practice. AI also has a growing role in auto-contouring, particularly for organs at risk, which streamlines the radiotherapy planning process and enables clinicians to focus more on complex treatment aspects. Recent studies have demonstrated the efficacy of AI-based auto-contouring in accurately delineating clinical target volumes (CTV) for rectal cancer, significantly reducing manual contouring time and variability, which highlights the practical benefits of AI integration into clinical workflows [114,115]. In practice, histology-based AI tools have already distinguished colorectal liver metastases from normal tissue with 99.5% accuracy, as reported by Kiritani et al. Moreover, Han et al. suggest that AI can predict histological growth patterns with AUC > 0.90. The clinicians of tomorrow should be excited for the prospect of tailored SABR doses based on aspects of tumour biology that we previously thought to be beyond the realm of modelling [116].

Additionally, the Royal College of Radiologists (2024) has published comprehensive auto-contouring guidance, emphasising structured assessments, clinician training, and rigorous quality assurance to ensure the safe and effective implementation of auto-contouring in clinical practice [117]. Applications of AI have been demonstrated in other types of radiotherapy, such as MR-linac, particularly in the treatment planning phases, optimising the delivery of radiotherapy [118].

Several reviews have highlighted the potential of AI in bioinformatics, particularly to identify new biomarkers for colorectal cancer, allowing for early diagnoses, reducing both morbidity and mortality [119,120]. Additionally, AI could have potential roles in pathology and endoscopy, to reduce the rate of misdiagnosis and improve diagnostic efficiency. For example, Cheng et al. crafted a transformer model using both H&E and IHC slides in colorectal cancer, achieving AUROC > 0.97 for MSI/MMRd status prediction. This work shows the efficiency at which AI can derive rich biomarkers and outcome information directly from pathological images [121]. Finally, AI has the potential to take into account all aspects of patient information, including biomarkers, genetic screening, radiology, and pathology, to generate personalised treatment plans for patients, with predictions of therapy response and survival. Algorithms can be trained on past patient data, and overall this would allow us to optimise treatment strategies and take advantage of patient-centred personalised medicine, maximising outcomes [120]. There are examples of studies that have used AI and machine learning for SBRT and colorectal cancer to predict response rates in both lung and liver metastases with some success [122,123]. The study using novel neural networks to assess liver SBRT survival had excellent success at predicting a negative outcome at the two-year threshold (100% success), with variable success at predicting a positive outcome at the threshold (56–82%). Overall, AI has a great potential in the future of SBRT and colorectal cancer; however, there are still key barriers to regular clinical application, including high-quality data to train algorithms, randomised controlled trials to prove that application provides a clinical benefit, technical/expert implementation of AI systems, and finally the significant cost associated with these processes. The era of AI is still in its youth, and with continual improvement and development it can strengthen clinical tools in diagnosis, prognosis, treatment planning, and outcome prediction.

## 3. Discussion

SABR has emerged as a transformative modality in the management of oligometastatic CRC, offering the potential for durable local control and prolonged survival in a subset of patients. As evidenced in multiple trials, including SABR-COMET and SABR-5, SABR provides compelling advantages in precision, toxicity profile, and long-term outcomes. However, despite its growing role, the treatment of oligometastatic CRC using SABR remains an evolving landscape—one marked by challenges in patient selection, tumour radioresistance, and integration with systemic and emerging therapies.

A key theme across the current literature is the variability in SABR outcomes depending on anatomical site, histological origin, and tumour genetics. CRC metastases, particularly those arising from rectal primaries or harbouring KRAS and TP53 mutations, demonstrate higher rates of local failure and a reduced responsiveness to standard SABR regimens. These findings underscore the urgent need for personalised dosing strategies, guided by genomic and anatomical biomarkers. The concept of Radiosensitivity Indices (RSIs) and site-specific BED10 thresholds is an emerging avenue with the potential to enhance tumour control while limiting toxicity. Furthermore, motion management remains a technical hurdle that limits the dosimetric precision of SABR. Innovations such as MR-guided SABR, real-time adaptive planning, and tumour trailing techniques have significantly improved target tracking and OAR sparing. These advances, while promising, are still subject to limitations like geometric distortion, increased complexity, and logistical constraints. Future studies are needed to refine these techniques and validate their clinical superiority, especially in CRC-specific cohorts.

In parallel, there is a growing recognition of SABR’s role within multimodal treatment paradigms. The integration of SABR with systemic therapy—particularly immunotherapy—has gained traction, especially given the potential for immune priming and the abscopal effect. Although most data to date stem from early-phase trials or extrapolations from other cancer types, the biological rationale for combining SABR with checkpoint inhibitors is strong. The early evidence suggests that SABR may modulate the tumour microenvironment to enhance T cell infiltration and sensitise microsatellite-stable CRC to immunotherapy, offering hope for otherwise resistant cases.

Looking ahead, AI represents a critical enabler of progress in this domain. AI and machine learning algorithms can optimise treatment planning, automate target contouring, and predict treatment responses by integrating genomic, radiomic, and clinical data. In particular, predictive models incorporating tumour genetics, volume, site, and prior therapies could help stratify patients into high- and low-benefit categories, informing dose escalation or adjunctive strategies like radiosensitisers. AI can also aid in synthesising complex multimodal data to tailor therapy, moving closer to the vision of precision SABR. However, the results remain suboptimal and are in their preliminary stages. More work must be performed to develop these models further.

Nonetheless, there remain substantial limitations in the evidence base. Few randomised phase III trials exist to confirm the long-term benefit of SABR over conventional approaches in CRC oligometastases. Heterogeneity in trial design, patient selection, and dose prescription further complicate interpretation and generalisability. Moreover, while early studies of SABR combined with immunotherapy or proton therapy are encouraging, larger and more focused studies are required to validate these strategies in CRC populations.

## 4. Conclusions

SABR is an effective local therapy tool for selected patients with oligometastic colorectal cancer. Still, outcomes are limited by biology (e.g., KRAS/TP53 mutations, rectal primaries) and site-specific radioresistance, particularly in the liver. Future progress relies on precision and integration: better patient selection using clinicogenomics, ctDNA, and radiomic/AI tools; technically sharper delivery via MR-guided adaptation, robust motion management, and selective proton use; and rational combination with systemic agents. The most pressing research gaps are setting up CRC-specific phase III trials, prospectively validated biomarkers to guide dose and timing, and defining BED thresholds tailored by site and genotype. Addressing these will shift SABR from being a broadly effective local treatment to a biologically guided, integrated strategy capable of improving survival outcomes for even more patients.

## Figures and Tables

**Figure 1 ijms-26-10302-f001:**
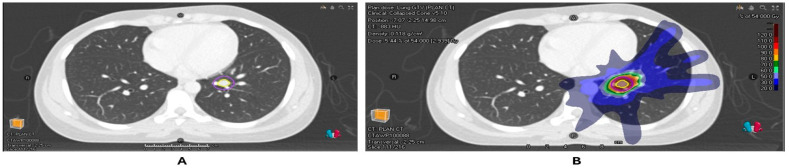
(**A**) Radiotherapy planning scan of patient with a colorectal cancer metastasis in the left lower lobe of the lung with GTV and PTV contours. (**B**) Dose distribution of the planned SABR of 54 Gy/3#.

**Figure 2 ijms-26-10302-f002:**
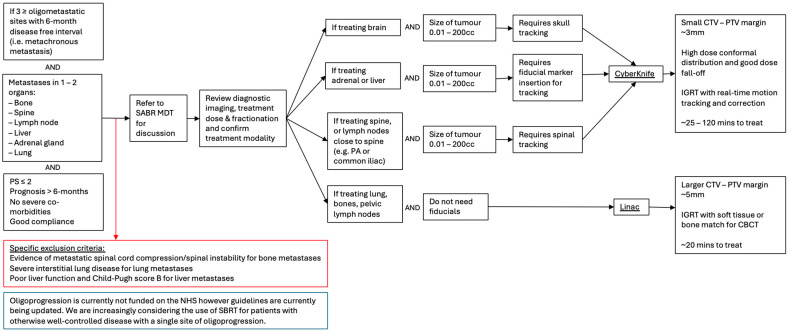
Flow chart outlining patient selection criteria for SABR in the management of colorectal cancer metastasis and MDT discussion of the appropriate treatment modalities.

**Table 2 ijms-26-10302-t002:** Key studies examining genomic alterations and their impact on SABR efficacy in colorectal cancer.

Study (First Author, Year)	Alteration(s) Examined	RT Setting (Site/Modality)	Reported Impact/Association	Key Results	Notes
Jethwa et al., 2020 [56]	*KRAS*; *TP53*; combined *KRAS + TP53*	Oligometastatic CRC, multi-site photon SABR (85 patients, 109 lesions)	*KRAS* mutations associated with reduced OS; *KRAS + TP53* double mutants with highest local-failure risk	1-year LF **44% vs. 11%** (WT); OS HR 2.4 for *KRAS*; HR **5.7** for *KRAS + TP53*	Retrospective; first dedicated genomic–SABR analysis in CRC metastases
Hong et al., 2017 [46]	*KRAS*; *TP53*; *KRAS + TP53*	Proton-based SABR, liver metastases (89 pts; 34 CRC)	*KRAS* mutation strongest predictor of poor LC; all double mutants failed locally	1 yr LC 42.9% (*KRAS* mut) vs. 72.1% (WT); *KRAS + TP53* 1 yr LC **20%**	Mixed primaries but CRC-dominant subset; modest dose range (30–50 GyE/5 fx)
Zhao et al., 2022 [72]	*KRAS* mutation status	Prospective phase II, lung and/or liver SABR for oligometastatic CRC (48 pts, 60 lesions)	*KRAS* status identified as an independent prognostic factor for local control, along with metastatic site, PTV, and time to metastasis	1 yr LC 88.3%, 3 yr LC 65.9%; median BED ≈ 100 Gy; no grade ≥ 3 toxicity	First prospective study to demonstrate *KRAS* status as a significant predictor of LC following SBRT in CRC oligometastases
Gui et al., 2021 [65]	*TP53* (driver) ± MYC-pathway	SRS, brain metastases from CRC (123 pts)	*TP53* drivers increases risk of intracranial progression; MYC alterations protective	HR 2.7 for intracranial progression with *TP53* driver; HR **0.15** for MYC-pathway	Stereotactic radiosurgery data but mechanistically relevant to SABR radioresistance
Wang et al., 2023 [59]	*dMMR/MSI-H*	Rectal chemoradiotherapy (locally advanced; 119 pts)	dMMR/MSI-H tumours relatively resistant to CRT; surgery-alone outcomes superior	Adjusted analysis: CRT arm worse OS and PFS in MSI-H cohort (*p* < 0.05)	Conventional RT, not SABR; included for context on genotype-specific radiosensitivity

Abbreviations: BED = biologically effective dose; CRC = colorectal cancer; CRT = chemoradiotherapy; LC = local control; LF = local failure; OS = overall survival; PFS = progression-free survival; SBRT/SABR = stereotactic body/ablative radiotherapy; SRS = stereotactic radiosurgery; WT = wild-type.

**Table 3 ijms-26-10302-t003:** Ongoing trials assessing the use of SABR and immunotherapy in colorectal cancer.

Trial Name	Trial Type	Progress of Trial	Population	Treatment
NCT06120127	Randomised controlled phase II	Recruiting	Colorectal liver metastases with high risk of local recurrence	Postoperative chemotherapy with/without radiotherapy and immunotherapy
NCT06603818	Prospective	Not yet recruiting	Microsatellite stable metastatic colorectal cancer	Tiragolumab and atezolizumab combined with radiotherapy
NCT06794086	Prospective, single-arm, phase II	Recruiting	Unresectable colorectal cancer liver metastases	SBRT combined with PD-1 inhibitors
NCT06130826	Phase I dose escalation study	Recruiting	Unresectable or metastatic colorectal cancer (and CEA positive breast cancer)	M5A-IL2 immunocytokine and SBRT
NCT06053996	Proof of concept phase II trial	Not yet recruiting	Metastatic colorectal cancer	Ablation of hepatic and pulmonary metastases with SBRT in combination with checkpoint inhibition
NCT06300463	Three-arm randomised phase II trial	Recruitment	Patients with colorectal cancer liver metastases pre-surgery	Combination immunotherapy (botensilimab/balstilimab) with or without radiation and/or TGFβ-CD73 trap
NCT06349044	Randomised multicentre phase II	Recruitment	Colorectal adenocarcinoma	SBRT, Probio-M9 microbial agents and PD-1 inhibitors

## Data Availability

No new data were created or analysed in this study. Data sharing is not applicable to this article.

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
