# Peer review of "Emerging Applications of Stereotactic Ablative Radiotherapy in Oligometastatic Colorectal Cancer"

_ijms, 2025, doi:10.3390/ijms262110302_

Round 1

Reviewer 1 Report

Comments and Suggestions for Authors

This review is well structured and covers an important topic: the role of SABR in oligometastatic colorectal cancer (CRC). The manuscript is generally clear and informative. It could be suitable for publication after some minor revisions.

1. In Abstract: “randomised data such as SABR-COMET, alongside large prospective series in- cluding SABR-5, have demonstrated improvements in survival outcomes and established SABR as a practical treatment option for patients with oligometastatic CRC” is overstated. Both SABR-COMET and SABR-5 included multiple cancer types, with only a small number of CRC patients. Please rephrase to indicate that survival benefit was observed in mixed populations and that CRC-specific evidence remains limited.

2. The immunotherapy section begins with a very positive statement. While the mechanistic rationale is described in detail, the clinical evidence in CRC is still very limited.  The study of AMP-224 showed no objective responses, and the atezolizumab + SBRT trial reported a median PFS of only 1.4 months, although a small subgroup of “elite responders” achieved longer benefit. Given these mixed results, it would improve the balance of the manuscript to moderate the introductory language and to emphasize more clearly that most CRC patients, especially those with MSS disease, currently derive limited benefit from SABR + immunotherapy, with responses confined to small subsets.
3. An error in the Funding section (“PThis”) and residual template text in the Author Contributions section.

Author Response

Reply is attached

Reviewer 2 Report

Comments and Suggestions for Authors

Comments

The manuscript titled “Emerging Applications of Stereotactic Ablative Radiotherapy in Oligometastatic Colorectal Cancer” offers a comprehensive and timely review of the role of SABR in the management of oligometastatic colorectal cancer. The authors effectively synthesize current evidence from key clinical trials, discuss technical and biological challenges, and explore emerging strategies such as MR-guided radiotherapy, proton therapy, immunotherapy combinations, and artificial intelligence. The review is well-structured and addresses a topic of significant clinical relevance, particularly as SABR becomes increasingly integrated into multimodal treatment paradigms. The inclusion of genomic, anatomical, and clinical factors influencing SABR outcomes adds depth and supports the theme of personalized therapy.

Overall, the manuscript is informative and well-referenced. However, several areas require improvement to enhance clarity, depth, and scholarly rigor. I recommend major revision before the manuscript can be considered for publication.

  1. The manuscript would benefit from a more critical synthesis of the evidence presented in Table 1. A summary paragraph comparing and contrasting the outcomes, limitations, and implications of these trials would add significant value.
  2. Figure 2 might be more reader-friendly if presented as a horizontal image.
  3. Some fonts in the text are not uniform, such as the first paragraph of 2.2.1. There are still some other format errors, such as one space should be left in front of the unit.
  4. The discussion on genomic factors (e.g., KRAS, TP53) is valuable but would be strengthened by including a summary table or figure illustrating the key mutations and their reported impact on SABR efficacy, similar to the approach used for clinical trials.
  5. The section on MR-guided SABR provides a good technical overview but could be improved by more directly linking these technical advantages to specific clinical outcomes in colorectal cancer patients.
  6. The description of the abscopal effect is interesting but somewhat disconnected from the main focus on oligometastatic CRC. The authors should more explicitly discuss the clinical relevance and realistic expectations for this phenomenon in the CRC population based on available evidence.
  7. The section on AI and machine learning is promising but currently speculative. Including more concrete examples of validated AI tools or ongoing trials specifically in colorectal cancer SABR would strengthen this section.
  8. The conclusion should more succinctly summarize the key messages and future directions, avoiding repetition of points already made in the discussion. It should also highlight the most pressing research gaps.

Author Response

Reply is attached

Round 2

Reviewer 2 Report

Comments and Suggestions for Authors

The authors have thoroughly and satisfactorily addressed all concerns raised in my previous review. The manuscript has been substantially improved through revisions to the abstract, the addition of tables, enhanced summaries, restructured organization, and refined language. The inclusion of relevant new content, together with a forward-looking conclusion, has significantly strengthened the overall quality and scholarly value of this review. I therefore recommend acceptance for publication.